# Retinal Macrophage-Like Cells as a Biomarker of Inflammation in Retinal Vein Occlusions

**DOI:** 10.3390/jcm11247470

**Published:** 2022-12-16

**Authors:** Dmitrii S. Maltsev, Alexei N. Kulikov, Yaroslava V. Volkova, Maria A. Burnasheva, Alexander S. Vasiliev

**Affiliations:** Department of Ophthalmology, Military Medical Academy, 21, Botkinskaya St, 194044 St. Petersburg, Russia

**Keywords:** macrophage-like cells, retinal vein occlusion, optical coherence tomography, macular edema, subfoveal fluid, optical coherence tomography angiography

## Abstract

Aim: To study the macrophage-like cells (MLC) of the inner retinal surface in eyes with retinal vein occlusions (RVO) and the association of MLC with clinical characteristics of RVO. Methods: In this retrospective cross-sectional study, the medical records and multimodal imaging data of treatment-naïve patients with unilateral RVO and no abnormalities of vitreoretinal interface electronic were reviewed and analyzed. To visualize MLC, structural projections of optical coherence tomography (OCT) angiography scans within a slab between two inner limiting membrane segmentation lines (with 0 and −9 µm offset) were evaluated. The density of MLC was calculated and compared between affected and fellow eyes of each patient with regards to OCT and clinical characteristics of RVO. Results: Thirty-six eyes (twenty-eight branch RVO and eight central RVO) of 36 patients (21 males and 15 females, mean age 48.9 ± 9.8 years) were included. The density of MLC in affected eye was statistically significantly higher than that of the fellow eye, 8.5 ± 5.5 and 4.0 ± 3.6 cells/mm^2^, respectively (*p* < 0.001). The MLC density in the affected eye had a statistically significantly correlation with that of the fellow eye (r = 0.76, *p* = 0.0001), but with none of the OCT and clinical characteristics of the affected eye apart from the presence of subfoveal fluid. Eyes with subfoveal fluid had a statistically significantly higher mean number of MLC than that of eyes without subfoveal fluid, 12.6 ± 6.3 and 6.9 ± 4.0 cells/mm^2^_,_ respectively (*p* = 0.009). Conclusion: The number of MLC on the inner retinal surface increases in RVO eyes which may reflect the activation of inflammatory pathways.

## 1. Introduction

Although retinal vein occlusion is a common retinal disorder affecting a significant proportion of the population, the pathophysiology of this condition is still not fully understood. Basic steps in the occlusion described by the Virchow’s triad include venous congestion, alteration of endothelial sheet, and blood hypercoagulability [1]. However, clinically significant complications of the RVO, namely macular edema and neovascularization, are driven by the changes in specific molecular signaling pathways.

From studies which analyzed vitreous samples, it is known that RVO is followed by the increase of intraocular concentrations of vascular endothelial growth factor (VEGF) and several proinflammatory cytokines controlling the integrity of the inner blood-retinal barrier [2,3]. This leads to an increase of vascular permeability, exudation, macular edema, and to further neovascular complications. Any study of the relationship between angiogenic and proinflammatory cytokines in RVO is therefore of high practical value as it will help to optimize the therapeutic approach. In cases where VEGF plays a leading role, patients benefit from anti-VEGF treatment. Where inflammatory reactions play a greater role, corticosteroids may be a more favorable approach [4].

Several studies have considered clinical biomarkers indicating the role of inflammation in RVO pathophysiology, including the types of fluid associated with macular edema, choroidal thickness, intraretinal hyperreflective foci, and anterior chamber flare [4,5,6]. However, several recent studies have described a new biomarker with the potential to characterize intraocular inflammation: macrophage-like cells of the inner limiting membrane [7,8]. With clinical OCT, these cells were defined as being evenly distributed over the macula spindle or star-like motile cells located above the inner limiting membrane surface. MLC demonstrated substantial changes in diabetic retinopathy, which is driven by ischemia and inflammation [9]. Since ischemia and inflammation also play an important role in RVO, we may expect some changes of MLC in RVO.

The aim of this study was to investigate changes of the density of MLC in eyes with RVO and the association of the density of MLC with the clinical and OCT characteristics of RVO.

## 2. Materials and Methods

This was a retrospective cross-sectional study. The study followed the ethical standards stated in the Declaration of Helsinki and was approved by the Local Ethics Committee. All participants signed informed consent for the use of their clinical data for investigation. Only treatment-naïve unilateral RVO patients were included in this study. Exclusion criteria were the presence of any abnormalities of vitreoretinal interface in either eye, including any stage of posterior vitreous detachment detected using optical coherence tomography (OCT), diabetes mellitus, glaucoma, history or presence of active intraocular inflammation, retinal vascular occlusion in the fellow eye, or any concurrent ocular condition impeding OCT imaging.

For all patients, electronic medical records and multimodal imaging data were reviewed and analyzed. All patients received OCT and OCT angiography (OCTA) (RTVue-XR, Optovue, Fremont, CA, software version 2017.1.0.150), green scanning laser ophthalmoscopy (F-10, NIDEK, Gamagory, Japan), fluorescein angiography (F-10, Nidek or Visucam 524, Carl Zeiss Meditec AG, Jena, Germany), and color fundus photography (AFC-330, NIDEK or Visucam 524, Carl Zeiss Meditec AG). All imaging procedures were performed after medically induced mydriasis. OCTA scans with a 6 × 6 mm field of view centered on the center of the fovea were used to obtain structural and vascular parameters, including the density of MLC, central retinal thickness (CRT), vessel density in superficial capillary plexus (SCP), and deep capillary plexus (DCP). The presence of subfoveal fluid was defined on structural scans crossing the central subfield as a hyporeflective space between the photoreceptor outer segment layer and the retinal pigment epithelium (RPE). The subfoveal choroidal thickness (SCT) was measured with the caliper tool as the distance from the RPE to the choroidal-scleral junction beneath the center of the fovea. The mean of three measurements was taken for analysis. Ischemic RVO was established if the area of retinal nonperfusion based on FA was larger than five and ten optic disc areas for BRVO and CRVO, respectively.

To visualize MLC, structural projections of OCTA scans within a slab between two inner limiting membrane segmentation lines (with 0 and −9 µm offset) were evaluated for both the affected and fellow eyes (Figure 1). With these settings, MLC were defined as small moderately reflective spots, some portion of which demonstrated a star-like or spindle-shape appearance. Using a cell counter tool, the density of MLC was calculated in ImageJ (NIH, Bethesda, CA, USA) by two experienced masked graders as the number of MLC per mm^2^. Correction of image magnification for myopic eyes was performed using Bennet’s formula before calculating MLC. For BRVO eyes, the density of MLC was calculated separately for the affected and unaffected areas. Based on multimodal imaging data, the area of RVO was defined as an area of retinal capillary hypo- or nonperfusion and/or accumulation of intraretinal fluid and/or intraretinal hemorrhages with or without cotton wool-spots. The RVO area was manually delineated on en face OCTA projections in ImageJ, measured, and converted to a mask which was further used to calculate MLC outside the affected area. The number of MLC within the affected area was calculated as the difference between the total number of MLC and the MLC number outside the affected area (Figure 2). Finally, the MLC density was calculated for the affected and unaffected areas of BRVO eyes.

Statistical analysis was performed in MedCalc 18.4.1 (MedCalc Software, Ostend, Belgium). The Kolmogorov–Smirnov test was used to check normality. The paired t-test was used to compare MLC density between the affected and fellow eyes of RVO patients as well as MLC density between the affected and unaffected area within BRVO eyes. The Wilcoxon test was used to compare MLC density between ischemic and non-ischemic RVO eyes as well as between central RVO (CRVO) and BRVO eyes. To define the factors associated with the density of MLC, the correlation coefficient was calculated for the density of MLC and age, clinical, retinal structural, and vascular parameters. Receiver operating characteristic (ROC) analysis was performed to evaluate MLC as a biomarker for the presence of subfoveal fluid. To assess the interrater repeatability of MLC density calculation, the intraclass correlation coefficient was calculated; *p* < 0.05 was considered statistically significant.

## 3. Results

Thirty-six eyes of 36 patients (21 males and 15 females, mean age 48.9 ± 9.8 years) were included. The mean LogMAR best-corrected visual acuity (BCVA) was 0.51 ± 0.27 (20/63 Snellen equivalent). The mean period after RVO onset was 25 days (ranging from 5 days to 3 months). There were twenty-eight BRVO and eight CRVO cases. Twelve BRVO and two CRVO cases were considered ischemic based on the FA data. The mean CRT and SCT was 466.3 ± 193.0 µm and 345.5 ± 105.4 µm, respectively. The mean area of the macula involved in RVO was 17.6 ± 9.8 mm^2^.

The density of MLC in the affected eye was statistically significantly higher than in the fellow eye, 8.5 ± 5.5 and 4.0 ± 3.6 cells/mm^2^, respectively (*p* < 0.001). This difference remained statistically significant in CRVO and BRVO separately (*p* < 0.05) (Figure 3). There was a strong correlation of MLC density between both eyes of each patient (r = 0.76, *p* < 0.001).

Although the density of MLC was higher in non-ischemic RVO compared to ischemic RVO eyes, this difference was not statistically significant (*p* = 0.38). The mean density of MLC within the affected area of BRVO eyes was statistically significantly lower compared to that of the unaffected region, 6.3 ± 5.3 and 10.5 ± 6.2 cells/mm^2^ (*p* = 0.009), respectively. The intraclass correlation coefficient values for MLC density in affected and fellow eyes were 0.94 (95% confidence interval (CI) 0.93–0.97) and 0.98 (95% CI 0.97–0.99), respectively. Age, time after occlusion, gender, BCVA, CRT, SCT, the area of the affected region, and vessel density in SCP or DCP had no association with MLC density in the affected eye (*p* > 0.05) (Table 1).

Twelve eyes (four CRVO and eight BRVO) had subfoveal fluid. Eyes with subfoveal fluid had statistically significantly higher density of MLC than the eyes without subfoveal fluid, 12.6 ± 6.3 and 6.9 ± 4.0 cells/mm^2^, respectively (*p* = 0.009) (Figure 4). ROC analysis showed the area under the ROC curve 0.83, sensitivity 88.9%, and specificity 66.7% for MLC as a biomarker for the presence of subfoveal fluid. Regarding the density of MLC in the fellow eye, there was a statistically significant difference between RVO eyes with and without subfoveal fluid, 5.8 ± 5.0 and 3.0 ± 2.3 cells/mm^2^, respectively (*p* = 0.04).

## 4. Discussion

In this study, we showed that MLC on the inner retinal surface more densely populate the macula in eyes with RVO compared to the healthy unaffected eyes of unilateral RVO patients. The density of MLC in RVO eyes varies significantly between individuals but has a strong correlation with the density of MLC in the fellow eye of each patient. Although these cells disappear from affected areas in eyes with BRVO, they seem to not be related to the ischemic status of RVO eyes. Moreover, apart from the presence of subfoveal fluid, MLC show no correlation with various clinical and OCT characteristics of RVO.

Retinal macrophage-like cells were first described on the inner limiting membrane using clinical OCT and image averaging by Castanos and coauthors [7]. This finding was later confirmed by adaptive optics imaging [8]. Both these studies demonstrated dendriform morphology and motility of the cells. Retinal cells, macrophages, glial cells, hyalocytes, and leukocytes may look like motile randomly distributed cells. However, leukocytes are normally absent in the healthy retina, while the cells on the inner limiting membrane are present in a high number in healthy eyes [7,8]. All other possible candidates for these cells have macrophage origin. It is possible, even without histopathological confirmation, to describe the cells under study as macrophage-like cells [9]. This term also allows us to define a particular cellular pool on the inner limiting membrane imaged with clinical OCT.

Changes of MLC in various posterior eye segment disorders have been proposed. However, MLC have only been studied in diabetic retinopathy where they demonstrated growth of the population during the conversion of non-proliferative retinopathy to PDR [10]. MLC have also shown increased density in the acute stage of multiple evanescent white dots syndrome [11]. Another feature registered in PDR was the accumulation of these cells along the large retinal vessels avoiding avascular regions. This may suggest the involvement of these cells in both vascular remodeling and in retinal ischemia. In our study, the density of MLC in the non-affected area was higher than in the affected area within the OCTA scan of BRVO patients. This phenomenon may indicate the migration of those cells from ischemia regions to normal areas, possibly due to the oxygen saturation gradient. Another explanation may be the recruitment of new MLC in uninvolved areas through the inflammatory pathways. The results of this study show that ischemia reduces the density of MLC. However, the ischemia in BRVO is a local factor, while inflammation is general (since the mediators of inflammation are distributed in the vitreous and affect the entire posterior eye pole). Therefore, MLC depopulation may be observed only over affected areas, while increase of MLC density may be seen over all non-involved regions. In general, the increase of MLC population appears to prevail over ischemic effects as is seen in CRVO cases, which demonstrate an increase of MLC population despite the absence of non-involved retina. We also cannot exclude that both migration and recruitment of MLC are operating simultaneously during RVO.

RVO are associated with overexpression, not only of VEGF, but also of many proinflammatory cytokines, including IL-6, IL-8, PIGF, MCP-1, ICAM-1, all of which increase retinal vessel permeability, leukocytes rolling and slow down the blood flow [3,6,12,13,14]. Activation of inflammatory signaling pathways may explain the relatively poor response to intravitreal anti-VEGF therapy in some cases of RVO. Such cases may benefit from corticosteroids therapy. However, identification of RVO cases where inflammation plays a leading role remains challenging. Inflammation biomarkers in RVO include subfoveal fluid and intraretinal hyperreflective foci [12,15], both of which have an association with MCP-1. MCP-1 is, in turn, responsible for the recruitment of monocytes and macrophages [16]. MCP-1 was shown to be significantly overexpressed in RVO. We therefore may expect activation of the MLC pool in this condition.

The problem with studying molecular signaling pathways in RVO mostly results from the invasive character of the procedures required to obtain aqueous humor or vitreous tape. Therefore, direct measuring of the intraocular level of different inflammatory factors in RVO is not appropriate outside of clinical studies. This highlights the importance of studying clinical biomarkers indicating the role of inflammation in each particular RVO case.

In this study, we found a significant increase of MLC population in eyes with RVO compared to fellow unaffected eyes. As was previously shown in eyes with diabetic retinopathy, MLC in RVO eyes avoided the retinal regions which have decreased perfusion or non-perfused areas. However, the density of MLC was still higher in RVO eyes. We may therefore conclude that MLC not only migrate from the area affected by the occlusion to unaffected areas, but also that some cells may be recruited to the inner retinal surface as was seen in multiple evanescent white dots syndrome [11]. The density of MLC does not correlate with the area of the occlusion or vessel density in SCP or DCP and therefore cannot be used as a biomarker for the ischemic status of the RVO. No other parameters of the RVO showed correlation with MLC density, including CRT, SCT or visual acuity. Only the presence of subfoveal fluid demonstrated an association with the density of MLC in RVO eyes. Subfoveal retinal fluid is a known biomarker of inflammation in RVO which was shown to be correlated with the levels of IL-6 and MCP-1 [12,17]. The high density of MLC in newly diagnosed RVO, taken together with other inflammatory biomarkers, may indicate the potential benefit of corticosteroids in the treatment of RVO. One confirmation for this suggestion can be obtained by monitoring changes in MLC density under the application of topical non-steroidal anti-inflammatory drugs.

Since the high density of the MLC in both affected and fellow eyes was associated with subfoveal fluid in RVO eyes and had a high interindividual difference, we suggest that baseline density of the MLC may indicate a predisposition to the activation of inflammatory pathways in RVO. In other words, if a patient had a high density of MLC, the inflammation may play a greater role in RVO, if any occurs in that patient. We suggest that these cells may participate in inflammatory pathways of RVO patients. Indeed, there are no known factors which could predict predominantly inflammatory status of RVO in each case. This status seems to be a unique characteristic of every affected eye, as is MLC density.

The limitations of these study are the strict exclusion criteria. Firstly, we avoided any cases with vitreoretinal interface abnormalities, including any stage of posterior vitreous detachment, since there are no data on the effects of changes of posterior vitreous on visualization of MLC. This resulted in inclusion of relatively young patients. The mean age of our study group was 49 years while the mean age of RVO patients in other studies is about 65 years. The effect of intraretinal hemorrhages and cotton-wool spots as factors distorting the retinal surface and potentially limiting MLC visualization also remains unknown. However, it would not change the main conclusion that MLC density increases in RVO and is associated with subfoveal fluid. Secondly, the consequence of applying strict inclusion criteria is the low number of cases included. Therefore, further studies with a larger and more diverse population of unilateral RVO patients is required. Thirdly, MLC of the foveal region were not identifiable on the structural en face OCT projections as was previously mentioned [7]. Finally, studies which measure vitreous and aqueous levels of proinflammatory mediators with regard to the density of MLC, as well as dynamic changes of MLC with the course of the disease are required. We did not include the control group in our study since interindividual difference in the fellow eyes of RVO patient was quite high (from 0.41 to 11.5 cells/mm^2^) and until now, there are no known factors defining the MLC density in healthy patients.

In conclusion, this study revealed the potential role of MLC of the inner retinal surface as a novel biomarker in RVO possibly indicating the activation of inflammatory pathways. MLC density increases in eyes with BRVO and CRVO and is associated with the accumulation of subfoveal fluid, a known biomarker of inflammation in RVO.

## Figures and Tables

**Figure 1 jcm-11-07470-f001:**
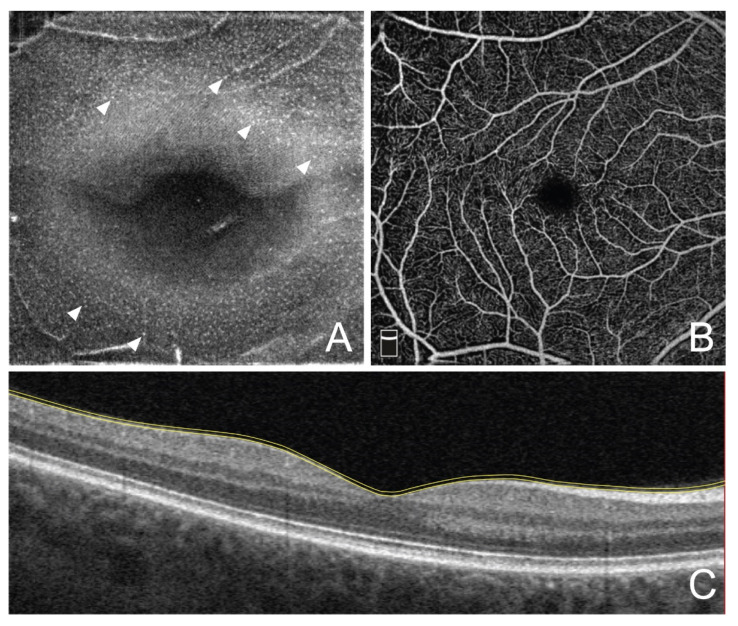
Representative example of visualization of retinal macrophage-like cells using 9-μm structural en face projection between two segmentation lines of the inner limiting membrane with −9 μm and 0 μm offset. (**A**) Structural en face optical coherence tomography angiography (OCTA) projection. Arrowheads indicate individual macrophage-like cells. (**B**) En face OCTA projection of superficial capillary plexus slab. (**C**) Cross-sectional OCT scan trough the center of the macula. Yellow lines represent segmentation lines of inner limiting membrane with 0 and −9 µm offset.

**Figure 2 jcm-11-07470-f002:**
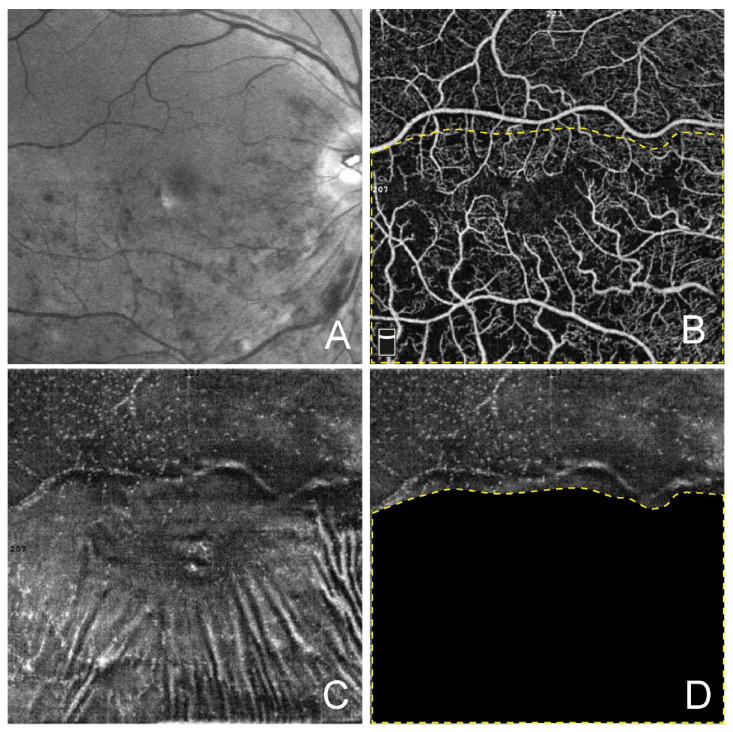
Counting of retinal macrophage-like cells in branch retinal vein occlusions. (**A**) Green scanning laser ophthalmoscopy showing area of occlusion. (**B**) En face optical coherence tomography angiography (OCTA) projection in the eye with branch retinal vein occlusion (BRVO). The yellow dashed line delineates the area of BRVO. (**C**) Structural en face OCTA projection displaying MLC. (**D**) Structural en face OCTA projection with the area affected by RVO masked to calculate MLC in unaffected area. The yellow dashed line delineates the area of BRVO.

**Figure 3 jcm-11-07470-f003:**
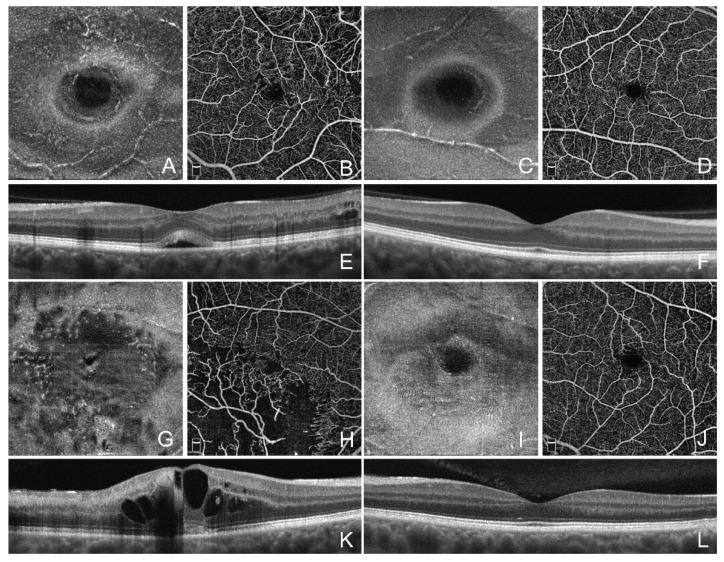
Optical coherence tomography angiography imaging of retinal macrophage-like cells in retinal vein occlusion with and without subfoveal fluid. (**A**) Structural en face optical coherence tomography angiography (OCTA) projection in the RVO eye with macular edema and subfoveal fluid. (**B**) En face OCTA projection of superficial capillary plexus slab. (**C**) Structural en face OCTA projection in the fellow unaffected eye. (**D**) En face OCTA projection of superficial capillary plexus slab in the fellow unaffected eye. (**E**) Cross-sectional OCT image showing macular edema and subfoveal fluid in the RVO eye. (**F**) Cross-sectional OCT image showing normal macula in the fellow eye. (**G**) Structural en face OCTA projection in the RVO eye with macular edema and no subfoveal fluid. (**H**) En face OCTA projection of superficial capillary plexus slab. (**I**) Structural en face OCTA projection in the fellow unaffected eye. (**J**) En face OCTA projection of superficial capillary plexus slab in the fellow unaffected eye. (**K**) Cross-sectional OCT image showing macular edema without subfoveal fluid in the RVO eye. (**L**) Cross-sectional OCT image showing normal macula in the fellow eye.

**Figure 4 jcm-11-07470-f004:**
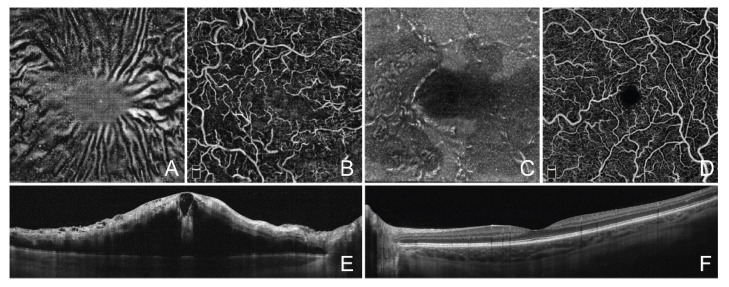
Optical coherence tomography angiography imaging of retinal macrophage-like cells in central retinal vein occlusion. (**A**) Structural en face optical coherence tomography angiography (OCTA) projection displaying macrophage-like cells. (**B**) En face OCTA projection of superficial capillary plexus slab. (**C**) Structural en face OCTA projection in fellow unaffected eye. (**D**) En face OCTA projection of superficial capillary plexus slab. (**E**) Cross-sectional OCT image showing severe macular edema in the affected eye. (**F**) Cross-sectional OCT image showing normal macula in the fellow eye.

**Table 1 jcm-11-07470-t001:** Correlation analysis for macrophage-like cells density and various parameters in eyes with retinal vein occlusion.

	MLC Density of RVO Eye	
	r	*p*-Value
Age	−0.13	0.53
Time after occlusion	0.14	0.55
BCVA	−0.21	0.11
CRT	0.09	0.68
SCT	0.02	0.92
Area of the affected region	0.12	0.55
Vessel density in SCP	−0.06	0.76
Vessel density in DCP	−0.03	0.87

BCVA, best-corrected visual acuity; CRT, central retinal thickness; DCP, deep capillary plexus; MLC, macrophage-like cells; RVO, retinal vein occlusion; SCP, superficial capillary plexus; SCT, subfoveal choroidal thickness.

## Data Availability

The primary data are available from Dmitrii S. Maltsev via email glaz.med@yandex.ru.

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
