# Peer review of "Retinal Macrophage-Like Cells as a Biomarker of Inflammation in Retinal Vein Occlusions"

_jcm, 2022, doi:10.3390/jcm11247470_

Round 1

Reviewer 1 Report

This is a relatively straightforward manuscript, describing observations using optical coherent tomography (OCT) that are consistent with elevation of the number and density of “macrophage-like cells” (MLCs) in the retinas of patients with documented unilateral retinal vein occlusion (RVO), compared to unaffected companion eyes.  The quality of the writing is good, as is that of the images presented.  Study design, analysis of data, and conclusions are reasonable.

Comments:

1-      The authors suggest that their findings are evidence for inflammatory involvement in RVO. So, how would this be informative for improved medical management of RVO patients?  Anti-inflammatory regimen?—and, if so, what agents specifically (e.g., daily aspirin? NSAIDs? Steroids?)?

2-      All of the quantitative data should be put into a Table, including the statistical parameters--- not just cited in the text.

3-       Minor edits:   be consistent-- either use capital P or lower case p to designate p-value.  For example: in the Abstract, see lines 22 vs. 25.  And elsewhere throughout the manuscript.

4-      Line 246, Data availability:  the term “data” is plural, so it should say “The primary data are available….”

Author Response

Dear Reviewer 1,

We would like to thank you for your time and effort in considering the manuscript for publication. Please find below our responses to your comments and tracking of changes.

Comment #1: The authors suggest that their findings are evidence for inflammatory involvement in RVO. So, how would this be informative for improved medical management of RVO patients?  Anti-inflammatory regimen?—and, if so, what agents specifically (e.g., daily aspirin? NSAIDs? Steroids?)?

Response: We suggest that this may help in choosing preferrable therapy for RVO associated macular edema between anti-VEGF drugs or steroids. We have pointed this out in the revised manuscript. We would like to thank the reviewer for the idea to study NSAID effects on MLC. This may help in clarification MLC as inflammatory biomarker in our future studies.

Lines: 232-236

Comment #2:   All of the quantitative data should be put into a Table, including the statistical parameters--- not just cited in the text.

Response: We have put all quantitative data of correlation analysis in Table 1 of the revised manuscript.

Lines: Table 1

Comment #3:       Minor edits:   be consistent-- either use capital P or lower case p to designate p-value.  For example: in the Abstract, see lines 22 vs. 25.  And elsewhere throughout the manuscript.

Response: In the revised manuscript we have turned all p values to capital ones.

Lines: 21, 24

Comment #4:      Line 246, Data availability:  the term “data” is plural, so it should say “The primary data are available….”

Response: Thank you for this note we have corrected this point throughout the manuscript.

Lines: 276, 249

Reviewer 2 Report

This research reported the macrophage-like cells (MLC) of the inner retinal surface in eyes with retinal vein occlusions (RVO) and association of MLC with clinical characteristics of RVO.

Please see below a few suggestions to help improve the clarity and impact:

I suggest revising and improving English language usage.

Line24: “that” should be “than”.

Line172: Spelling error, “leukocytes”

1.     Line173 to Line 175:Please clarify the expression of this statement.There may be some Grammatical mistakes make it difficult to be understood.

2.     I think the definition of “Retinal Macrophage-Like Cells” should be added in the introduction section as it’s the major research object of this study.

3.     It will be helpful to find a cut-off value and report its sensitivity and specificity if you want to make MLC a biomarker of Inflammation in RVO.

4.     Please review how the macrophage like-cells “also play an important role in RVO”?

5.     Please clarify background demographics of the patients: affected area, macular edema or involvement, and ischemic involvement.

6.     Please recorrect the reference format. Should the 4 be ref [4] in line 45, and also in line 51?

7.     These macrophage-like cells on the foveal ILM surface of the macula were unable to visualize due to their special morphology. Please clarify the method of image analysis to improve cell identification.

8.     Please describe details of identification and isolation of MLC layer from OCT en face plate in the method part.

9.     Perhaps a brief description of the classification between ischemic and nonischemic should be included.

10.  Please consider to separately evaluate MLC densities by two masked observers to assess inter-rater reliability.

Author Response

Dear Reviewer 2,

We would like to thank you for your time and effort in considering the manuscript for publication. Please find below our responses to your comments and tracking of changes.

Comment #1: I suggest revising and improving English language usage. Line24: “that” should be “than”. Line172: Spelling error, “leukocytes”

Response: We have corrected spelling in these points as well as throughout the manuscript.

Lines: 190

Comment #2: Line173 to Line 175:Please clarify the expression of this statement.There may be some Grammatical mistakes make it difficult to be understood.

Response: We have modified this phrase to clarify the meaning.

Lines: 191-193

Comment #3: I think the definition of “Retinal Macrophage-Like Cells” should be added in the introduction section as it’s the major research object of this study.

Response: We have provided the definition for MLC in the introduction section.

Lines: 51-52

Comment # 4: It will be helpful to find a cut-off value and report its sensitivity and specificity if you want to make MLC a biomarker of Inflammation in RVO.

Rsponse: We have performed the ROC analysis to define sensitivity and specificity of MLC as a biomarker for SRF appearance in RVO, which showed AUROC = 0.83, Sensitivity = 88.9%, and Specificity = 66.7%. This was noted in the revised version of the manuscript in the material and methods section and in the results.

Lines: 120-121, 164-166

Comment #5: Please review how the macrophage like-cells “also play an important role in RVO”?

Response: This phrase is referred to inflammation and ischemia which “also play an important role in RVO” as in DR. We have corrected this phrase to avoid misunderstanding. Thank you.

Lines: 52-55

Comment #6: Please clarify background demographics of the patients: affected area, macular edema or involvement, and ischemic involvement.

Response: In the revised manuscript we have provided an information on the area affected by the occlusion by the mean of OCTA, severity of macular edema as CRT value, and ischemic involvement as the proportion of ischemic and non-ischemic occlusions.

Lines: 129-131

Comment #7: Please recorrect the reference format. Should the 4 be ref [4] in line 45, and also in line 51?

Response: We have corrected reference format where it was required. Thank you!

Lines: 45, 54

Comment #8: These macrophage-like cells on the foveal ILM surface of the macula were unable to visualize due to their special morphology. Please clarify the method of image analysis to improve cell identification.

Response: Indeed, MLC of the foveal region were not identifiable on the structural en face OCT images as it was mentioned in a previous paper by Castanos et al. This might be related to the specific morphology of retinal microglia  in this region, or reflect actual lack of these cells in this region, or an effect of foveal morphology (foveal slope) on the OCT imaging of those cells. Anyway, until now there is no method for clear imaging of those cells in the foveal region, except glial cells in the middle of the FAZ which were not taken into the analysis in our study. In the revised manuscript we have mentioned this fact in the limitation section.

Lines: 254-256

Comment #9: Please describe details of identification and isolation of MLC layer from OCT en face plate in the method part.

Response: We have provided the details on the imaging and identification of MLC with structural enface OCT in the revised version of the manuscript.

Lines: 88-90

Comment #10:    Perhaps a brief description of the classification between ischemic and nonischemic should be included.

Response: In the revised version of the manuscript, we have provided the description for classification between ischemic and nonischemic RVO.

Lines: 83-85

Comment #11:  Please consider to separately evaluate MLC densities by two masked observers to assess inter-rater reliability.

Response: Inter-rater reliability for MLC counting in affected and fellow eyes of RVO patients is presented in lines 153-154.

Lines: 153-154

Round 2

Reviewer 2 Report

It is a good research direction for the author to apply the research method of diabetes retinopathy to retinal vein occlusion. I propose the following questions.

1. In the OCTA scanning area of BRVO patients, how are the affected areas and non-affected areas divided? What is the standard of “the yellow dashed line”( in Figure 2)? Please list references.

2. The article points out that the density of MLC in the non-affected area is higher than that in the affected area in the scanning area of BRVO patients. Please explain this phenomenon.

3. The conclusion of the article is that MLC density in RVO patients' eyes is higher than that in the contralateral eyes, indicating that it is an indicator of inflammatory activity. However, the article also said: MLC density in the non-affected area of BRVO patients is higher than that in the affected area. Can we conclude that the inflammation activity in the non-affected area of BRVO patients is higher? This obviously does not conform to common sense. Please explain.

4. It is mentioned that “Eyes with sub foveal fluid had statistically significantly higher density of MLC than the eyes without sub foveal fluid”, but “Age, time after occlusion, gender, BCVA, CRT, SCT, the area of the affected region, vessel density in SCP or DCP had no association with MLC density in the affected eye (P > 0.05)”. Can it be understood that the increase of MLC density can improve the permeability of blood vessels in the retina? So why does the increase of MLC density in the non-affected area of the BRVO affected eye not cause retinal vascular leakage or sub foveal fluid accumulation? In conclusion, in BRVO patients, the increase of MLC density in the unaffected area is a very confusing problem, which will bring a lot of incomprehensible content to the article.

5. It is suggested that the baseline MLC density of patients may predict the inflammatory response level of RVO patients. In this case, why is there no normal control group in this study? Only the follow eyes of the patients were set as the control. In addition, what is the classification standard of inflammatory response level of RVO patients? Is it just a sub foveal fluid? This conclusion is obviously somewhat exaggerated. The article only proves that there is a correlation between the density of MLC and sub foveal fluid.

6. In addition, this study is a cross-sectional study. The scanning image of OCTA is greatly affected by retinal edema and hemorrhage. At each follow-up, macular edema and bleeding may be different, so will the scanning results be very different? How repeatable are the results of this cross-sectional study? In addition, the sample size of patients also needs to be increased.

Author Response

Dear Reviewer,

We would like to thank you for your time and effort in considering the manuscript for publication. Please find below our responses to your comments and tracking of changes.

Comment #1. In the OCTA scanning area of BRVO patients, how are the affected areas and non-affected areas divided? What is the standard of “the yellow dashed line”( in Figure 2)? Please list references.

Response: In the revised version of the manuscript, we have provided a definition for the RVO area.

Lines: 94-97

Comment #2. The article points out that the density of MLC in the non-affected area is higher than that in the affected area in the scanning area of BRVO patients. Please explain this phenomenon.

Response: Thank you. We have added this point to the discussion section of the revised manuscript. We believe that two mechanisms may play a role: MLC may migrate from ischemic areas to unaffected ones or MLC may be attracted from blood in unaffected areas through the inflammatory pathways.

Lines: 208-212

Comment #3. The conclusion of the article is that MLC density in RVO patients' eyes is higher than that in the contralateral eyes, indicating that it is an indicator of inflammatory activity. However, the article also said: MLC density in the non-affected area of BRVO patients is higher than that in the affected area. Can we conclude that the inflammation activity in the non-affected area of BRVO patients is higher? This obviously does not conform to common sense. Please explain.

Response: Thank you for this note. Indeed, these two facts seem to be controversial. However, in RVO eyes as in DR eyes there are at least two main driving forces: ischemia and inflammation. Our results show that ischemia reduces the density of MLC while inflammation increases their density. The ischemia in BRVO is a local factor while inflammation is general (since the mediators of inflammation are distributed in the vitreous and affect entire posterior eye pole). Therefore, from our point of view inflammatory response prevails over ischemic effects. CRVO cases which demonstrate similar changes in MLC density confirm this suggestion.

Lines: 208-220

Comment #4. It is mentioned that “Eyes with sub foveal fluid had statistically significantly higher density of MLC than the eyes without sub foveal fluid”, but “Age, time after occlusion, gender, BCVA, CRT, SCT, the area of the affected region, vessel density in SCP or DCP had no association with MLC density in the affected eye (P > 0.05)”. Can it be understood that the increase of MLC density can improve the permeability of blood vessels in the retina? So why does the increase of MLC density in the non-affected area of the BRVO affected eye not cause retinal vascular leakage or sub foveal fluid accumulation? In conclusion, in BRVO patients, the increase of MLC density in the unaffected area is a very confusing problem, which will bring a lot of incomprehensible content to the article.

Response: We theorize that the correlation between MLC density and SRF may not have causative relationship but rather may be explained through the action of inflammatory pathways similar to the increased concentration of proinflammatory mediators in BRVO eyes which do not result in generalized retinal edema. In the revised manuscript we have additionally discussed the difference in MLC population between affected and unaffected areas and provided a plausible explanation.

Lines: 208-220

Comment #5. It is suggested that the baseline MLC density of patients may predict the inflammatory response level of RVO patients. In this case, why is there no normal control group in this study? Only the follow eyes of the patients were set as the control. In addition, what is the classification standard of inflammatory response level of RVO patients? Is it just a sub foveal fluid? This conclusion is obviously somewhat exaggerated. The article only proves that there is a correlation between the density of MLC and sub foveal fluid.

Response: The problem of adding a control group is related to the high interindividual difference in MLC density in healthy eyes. Even within our study fellow eyes showed the range of MLC density 0.41 to 11.5 cells/mm2. Since MLC studying is a new topic the predictors for individual MLC density are not known yet. At the same time, SRF is a known biomarker of high levels of proinflammatory cytokines in RVO.

We agree that the final statement on the role of MLC as a biomarker for inflammation in RVO is somewhat speculative and we have modified the conclusion in the revised version of the manuscript accordingly.

Lines: 275-278, 279-282

Comment #6. In addition, this study is a cross-sectional study. The scanning image of OCTA is greatly affected by retinal edema and hemorrhage. At each follow-up, macular edema and bleeding may be different, so will the scanning results be very different? How repeatable are the results of this cross-sectional study? In addition, the sample size of patients also needs to be increased.

Response: We agree with the reviewer that macular edema may affect the imaging of MLC and therefore we only included patients with normal vitreoretinal interface. This was highlighted in both the abstract and the main manuscript. This study did not analyze changes of MLC density over time and any changes of macular edema could not affect our results.

Lines: 13, 64-65